# Effect of high-dose β-Alanine supplementation on uphill cycling performance in World Tour cyclists: A randomised controlled trial

**Silvia Pérez-Piñero**[1], **Domingo Jesús Ramos-Campo**[2], **Francisco Javier López-Román**[1,3]*, **Raquel Ortolano**[1], **Antonio Torregrosa-García**[1], **Antonio Jesús Luque-Rubia**[1], **Natalia Ibáñez-Soroa**[1], **Luis Andreu-Caravaca**[1,4], **Vicente Ávila-Gandía**[1]

1 Faculty of Health Sciences, Sports Physiology Department, UCAM Universidad Católica San Antonio de Murcia, Murcia, Spain, 2 Faculty of Physical Activity and Sport Science-INEF, Department of Health and Human Performance, LFE Research Group, Universidad Politécnica de Madrid, Madrid, Spain, 3 Biomedical Research Institute of Murcia (IMIB-Arrixaca), Primary Care Research Group, Murcia, Spain, 4 Facultad de Deporte, UCAM, Universidad Católica de Murcia, Murcia, Spain

* jlroman@ucam.edu

**Data Availability Statement:** The data underlying the results presented in the study are available upon request to Catholic University of Murcia's

## Abstract

Scientists and coaches seek effective ergogenic aids for performance improvement. Cyclists commonly use β-Alanine, which may enhance post-exercise recovery and physical performance. High-dose β-Alanine supplementation's impact on World Tour cyclists during a 7-day camp remains unstudied. This study aimed to analyse the effect of a high dose of β-alanine in World Tour cyclist during a 7-day camp. A double-blinded, randomised controlled trial was conducted. 11 cyclists were included in the final analysis: β-alanine supplementation (n = 5; $VO_2$max: 67.6±1.6 ml/kg/min) and a placebo group (n = 6; VO2max: 68.0±2.4 ml/kg/min). The duration of the supplementation protocol was seven days with four daily intakes. The subjects commenced supplementation after the physical tests (immediately following the snack) and consumed the final intake after breakfast on the day of the final test (a total of 7 days and 3 additional doses, 31 servings in total; 5g per dosage; 155g the total cumulative amount). Before and after seven days of supplementation, the cyclists performed an uphill time trial. Blood lactate, heart rate and rating of perceived exertion were measured during test. β-alanine supplementation improved the relative mean power attained during the time-trial compared with the control group (Z = -2.008; *p* = 0.045; Δ = 0.060), as well as the time needed to complete this trial (Z = -2.373; p = 0.018). As for physiological and metabolic variables, no significant change was found. In conclusion, the present study supports the effectiveness of one-week high dose of β-alanine during a cycling training in World Tour cyclists to improve their uphill time-trial performance. In addition, it is important to highlight the potential role of β-alanine in improving recovery power. This aspect is particularly relevant in the context of a training camp, where fatigue levels can increase alongside training intensity.

   **Trial registration**: This study was registered in ClinicalTrials.gov: (identifier: NCT04427319).

Science Ethics Committee. Dra. Lourdes Meroño García; lmerono@ucam.edu.

**Funding:** The author(s) received no specific funding for this work.

**Competing interests:** The authors have declared that no competing interests exist.

## Introduction

Professional road cycling performance is a complex subject that is affected by a number of uncontrollable variables, such as altitude, tactic, wind, weather and so on [1]. From a physiological point of view, aerobic metabolism is prevalent in professional cycling. However, anaerobic metabolism determines performance in some phases of the stages, such as sprint, mountain stages or time-trial stages [1]. In addition, the ability to maintain high maximal mean power outputs after high amounts of work done (i.e., fatigue) is an important parameter for success in professional cyclists [2]. During these segments, an increased reliance on glycolysis to maintain ATP supply indicates a greater anaerobic energy release [3]. During the intense muscle activity that is associated with these efforts, an increase in cellular production of $H^+$ greatly contributes to fatigue by muscle acidosis [4]. This could eventually reduce power during high-intensity phases, such as in mountain ascents. Thus, increasing buffering capacity could help to improve performance in these high-intensity efforts during a cycling race.

Developing the most effective and efficient training method to optimize cycling performance has been the focus of many scientists and coaches. While training strategies are crucial, it's also important to consider the integration of legal ergogenic aids, which is a popular approach among athletes, particularly cyclists, seeking to maximize their physical performance [5]. Among others, intake β-alanine (β-alanine) is used by high-level athletes and specifically by cyclists, and is one of the five currently most popular supplements endorsed by International Olympic Committee [6]. β-alanine is a nonproteogenic amino acid that once ingested combines with L-histidine within skeletal muscle and other organs to form carnosine [7], which is the rate-limiting step in muscle carnosine synthesis [8]. The increase of carnosine content in skeletal muscle enhances intracellular buffering capacity, which enables a greater tolerance of sustained anaerobic activity [7]. Previous studies have also suggested that intramuscular carnosine can increase muscle force production due to the higher $Ca^{+2}/H^+$ exchanger in the fibre, increased $H^+$ binding to carnosine and $Ca^{+2}$ unloading at the sarcomere, subsequently increasing cross-bridge formation and force production [9,10]. Moreover, the antioxidant, antiglycating and ion-chelating effects of carnosine have been reported [11–13]. This suggests the possible beneficial effect of β-alanine supplementation in the recovery process and subsequent exercise performance [7].

The existing evidence from previous meta-analysis studies has demonstrated the efficacy of β-alanine supplementation on exercise capacity and performance [14,15]. However, some moderating factors can alter the effectivity of the supplementation protocol. In particular, greater gains are expected in exercises from 30 s to 10 min of duration, while other shorter (<30 s) [14] or in longer exercise durations (i.e. physical performance in aerobic–anaerobic transition zones) [16] show no effect or controversial results. Most previous studies of the effect of β-alanine supplementation on cycling performance have been developed in laboratory conditions [17–19], which is far from a practical point of view. Recently, a study analysed the effect of β-alanine supplementation on cycling performance using a novel exercise protocol that is representative of the demands of road-race cycling but in laboratory conditions. The authors showed that chronic β-Alanine supplementation increased muscle carnosine content but did not improve cycling performance [20].

Study population can also affect the result of β-alanine supplementation because the effect of β-alanine on trained individuals is lower than in nontrained individuals [14]. Consequently, the background of β-alanine in cycling has been developed using amateur or trained cyclists [18–20]. The effect in World Tour cyclists is scarce. The only study was published by our research group, which analysed the effect of β-alanine supplementation in this population [21].

Finally, the β-alanine effect is also modified by supplementation dosage. The efficacy of chronic supplementation with doses ranging from 1.5 to 6.4 g/day for between 4 and 10 weeks has been shown for athletic performance [7,14]. However, doses higher than 6.4 g per day produce a greater risk of symptoms of paresthesia. This sensation of flushing is associated with an irritant tingling in the ears, scalp, hands, and torso [16] and is the limiting factor for acute B-ala intake. Recently, to avoid this symptom, the supplementation protocol has been focused on administering smaller amounts of a sustained-release formulation of β-alanine throughout the day, which permits greater doses without the risk of paresthesia. For example, a previous study has used higher dosing protocols (12 g per day) of sustained-release formulation to accelerate the increase in carnosine content in skeletal muscle while attenuating paresthesia [22]. However, only one study by our research group has used higher dosage of β-alanine (20 g/day) during seven days of supplementation to optimise cycling performance without paresthesia [21]. No study has yet analysed the effect of high-dose β-alanine supplementation during seven days in a real uphill cycling time trial in World Tour cyclist. Therefore, the aim of the present study was to analyse the effect of one-week high dose of β-alanine during a cycling training camp that was held for World Tour cyclists to improve their uphill time-trial performance. Based on the previous research, our hypothesis was that β-alanine supplementation would significantly improve power and uphill time-trial performance in World Tour cyclists.

## Methods

### Design

A double-blind, randomised controlled trial (RCT) was conducted with two arms—that is, β-alanine supplementation (BA) and a placebo group (PLA)—during one intensive training camp week (seven days) (Fig 1). The research was performed during the first camp of the season (January). To analyse the effect of β-alanine on uphill cycling performance, an uphill time-trial measurement was carried out at pre- and post-7-days of β-alanine supplementation (BA) or placebo (PLA). Simple randomization was performed by a scientist not participating in the study, using software (Epidat 4.2, Galicia, Spain) that generated random codes which were assigned to participants. The trial was approved by the Ethics Committee of UCAM (ID number: CEO12004) and was conducted in accordance with the Declaration of Helsinki. The trial design followed Consort guidelines for RCT. This study was registered in ClinicalTrials.gov (identifier: NCT04427319).

### Participants

In total, 12 World Tour cyclists belonging to a top five road World Team of the Union Cycliste Internationale (UCI) were initially recruited for this study. In total, 11 were included in the final analysis because one participant from the BA group dropped out from the study due to a race that he had to attend. Recruitment began on March 6 and ended on June 12, 2020. The sample size was calculated according to the mean power as the main variable of the study. Considering a standard deviation of mean power of 12.3 W reported in a similar population [21], for a precision of 17 W with an alpha risk of 5% and statistical power of 80%, for a comparison means test (t-test), 6 subjects in each group. Cyclist were included if they were male professional active cyclist belonging to a UCI World Team and starting the study in a well-rested condition (i.e., not performing intense training the day before). However, they were excluded if: i) they had a chronic disease; ii) had an injury during the month before the study that can affect their training load; iii) had an allergy to β-alanine or placebo components; iv) ingested β-alanine supplementation in the two months before the study; and v) used other

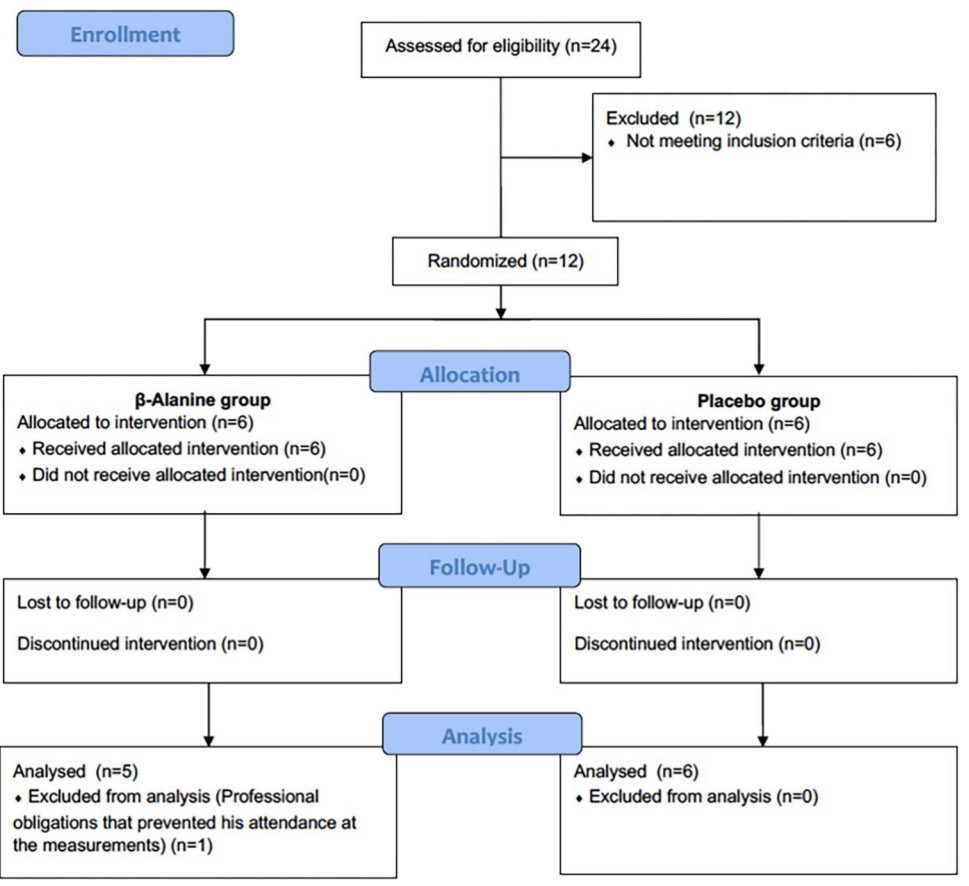

**Fig 1. CONSORT diagram of the study.**

performance enhancer supplements during the study or 15 days before it. All of the participants read and signed an informed consent before starting the study.

## Supplementation protocol

Cyclists were randomly allocated to a sustained-release β-Alanine powder blend (BA; BETA-FOR3MAX®, Martinez Nieto S.A., Cartagena, Spain) or placebo (PL; uncooked wheat semolina from Triticum durum) group. Participants started supplementation after the first testing session and finished it after the breakfast of the same day of the post testing session. In total, the supplementation protocol lasted for seven days with four daily intakes (i.e., breakfast, lunch, snack and dinner). Each single dosage consisted of: 5 g of β-alanine 37.5 mg of L-histidine and 12.5 mg of carnosine (20 g of β-alanine,187.5 mg of L-histidine and 62.5 mg of carnosine per day). A total of 31 servings of 155 g of β-alanine, 1.45 g of L-histidine and 96.87 mg of carnosine (the total cumulative amounts of supplementation) were ingested by the cyclist. None of doping substance were found in the batch employed (PI080120NID), which was analysed by an external laboratory (Informed Sports LGS Supplement Screen service, LGC Group, Cambridgeshire, United Kingdom; certificate of analysis number 3454). The placebo was manufactured by the same company. Both products were similar in appearance and stored in opaque containers. They were weighed and the total amount consumed was checked for compliance. The placebo consisted in 33 g of semolina per day—energy: 126 Kcal, protein: 3.8 g, carbohydrate: 23.1 g, of which sugar were only traces, fibre: 1.3 g, fat: 0.2 g. Additionally, to

check blinding of participants, cyclists were surveyed about their beliefs on which product they had received after the study.

## Uphill time-trial test

After a standardized warm up (22 minutes following the competitive warm up strategy and power zones of each cyclist: 3 min easy spin + 3 min endurance + 3 min tempo + 2 min at functional threshold power + 2 min easy + 1 min maximum oxygen uptake + 2 min easy + 3 min activation (10 s on + 50 s off) + 3 min easy), the cyclist performed an uphill time trial in the Coll de Rates (Alicante, Spain) over a distance of 4.5 km and with a slope of 5%. All the cyclists used the same model of bicycle (Shimano Ultegra Di2, Eindhoven, Netherlands), which is the one provided by the team for the season. All the bicycles were adapted to each athlete with biomechanical studies. During the test, mean power (absolute (W) and relative (W/kg)) was measured using a powermeter (Garmin Vector 3, Garmin International, Olathe, KS, USA). In addition, the time to complete the time trial was registered. Additionality, a capillary blood sample (5 µl) was collected by a finger prick to determine blood lactate concentration using the Lactate Pro 2 (Lactate Pro, Arkay, Inc, Kyoto, Japan) pre-, immediately post- and 3 min post-exercise. The mean heart rate (Polar RS800, Polar Electro Oy, Kempele, Finland) was recorded during the time trial. In addition, ratings of perceived exertion (RPE) were determined using the 10-point Borg scale following warm up and after the time trial. A questionnaire to analyse paresthesia [21] was used in baseline, after the first day of intake and after the last day of intake.

## Training program during the training camp

All of the cyclists involved in the experiment performed the same training plan during the week. All of the training sessions were recorded using GPS bike computer (Garmin Edge 530, Garmin International, Olathe, KS, USA) and a powermeter (Garmin Vector 3, Garmin International, Olathe, KS, USA). The training plan included two days of low intensity cycling workouts and three days of moderate intensity with uphill workout. Cycling was performed during the training session together and at the same time of the day. Training load was calculated as the daily mean during the five days of intense training camp week. The first and the last day (testing days) were not included in the training load. External load was measured using the Training Stress Score™ (TSS) proposed by Coggan [23] using the formula:

$$TSS = [(t \times NP \times IF)/(FTP \times 3600)] \times 100$$

where t is the duration of the workout, NP is normalized power, IF is a ration between NP of the workout and the individual's functional threshold power (FTP), which give an intensity factor. FTP was assessed by an incremental lab test based on the power output at 4 mmol/L of lactate. The test commenced with an initial resistance of 110 W, with subsequent increments of 35 W every 4 minutes [24].

## Statistical analysis

Data collection, treatment and analysis were analysed using the SPSS for Windows statistical package (version 25.0; IBM, Armonk, NY). First, descriptive statistics with measures of central tendency (median) and dispersion (interquartile range (IQR)) were calculated. Taking into account the sample size, non-parametric tests were implemented. Initially, we assessed whether there were statistically significant improvements in each variable from the initial to the final measurement within each group. For this purpose, we employed the Wilcoxon Signed-Rank Test. To determine the efficacy of the product, we calculated the changes in each

variable (final test value—initial test value) and compared these changes between the experimental and placebo groups using the Mann-Whitney U Test. Additionally, the difference in increments between the placebo and product groups is reported. For all procedures, a level of significance of p≤0.05 was selected to indicate statistical significance.

## Results

In total, 12 world tour cyclists were initially recruited and 11 were included in the final analysis. One subject from the BA group dropped out from the study due to a race that he had to attend. The cyclists were allocated in PLA and BA. Descriptive characteristics are presented in Table 1. No significant differences between groups were found in these descriptive characteristics of the sample.

With respect to the performance variables, a significant effect was observed on the time trial (Z = -2.373; $p$ = 0.018; Δ = -10.00; Δ IQR: -10.0–25.0), showing an improvement in BA in the post-test (Pre: 606.6; IQR: 592.8–633.5 s; Post: 588.0; IQR: 540.0–620.0 s; $p$ = 0.043; Z = -2.023) in comparison with baseline, while PLA maintained the time trial in both testing session (Pre: 620.0; IQR: 609.3–637.8 s; Post: 637.0; IQR: 613.8–659.8 s; $p$ = 0.249; Z = -1.153) (Fig 2). Similarly, a significant effect was observed in the relative power developed during the test (Z = -2.008; $p$ = 0.045; Δ = 0.060; Δ IQR: -0.13–0.11). This difference can be attributed to the non-significant improvement observed in BA (Pre: 6.05; IQR: 5.7–6.1 W/kg; Post: 6.1; IQR: 5.8–6.3 W/kg; $p$ = 0.043; Z = -2.023) and the maintenance in PLA (Pre: 5.77; IQR: 5.6–5.9 W/kg; Post: 5.65; IQR: 5.5–5.9 W/kg; $p$ = 0.249; Z = -1.153) (Fig 3). In addition, no main effect was found in absolute power (Z = -1.643; $p$ = 0.100; Δ = 7.12; Δ IQR: -15.83–22.58), indicating maintenance in PLA (Pre = 438.19; IQR: 413.6–477.8 W; Post: 447.8; IQR: 399.5–460.9 W; $p$ = 0.600; Z = -0.524) and a non-significant improvement in BA (Pre = 456.3; IQR: 417.3–463.8 W; Post: 469.9; IQR: 433.1–494.8 W; $p$ = 0.080; Z = -1.753).

Regarding physiological and metabolic variables, no significant main effect was found in mean heart rate (Z = -1.643; $p$ = 0.100; Δ = 0.60; Δ IQR: -6.22–4.17) (Fig 4A) and in blood lactate concentration (Z = -1.555; $p$ = 0.120; Δ = 0.200; Δ IQR: -2.00–0.30) (Fig 4B). Moreover, no significant effect was observed in the rating of perceived exertion (Z = -2.220; $p$ = 0.026; Δ = 0.00; Δ IQR: -0.00–1.00) (Figs 4C, 5 and 6).

Finally, Table 2 shows the training load characteristics of the five days of intense training (the test days was not included). No differences between groups were observed in volume (hours: $p$ = 0.766), normalized power ($p$ = 0.948) or training stress score ($p$ = 0.517).

## Discussion

To our knowledge, this is the first study to analyse the effect of one-week high dose of β-alanine during a cycling training camp that was held for World Tour cyclists to improve their uphill time-trial performance. The main finding of the present research is that β-alanine supplementation improved cycling uphill time-trial performance by relative mean power and

**Table 1. Subject characteristics of the sample.**

| Characteristics | PL (n = 6) | BA (n = 5) |
|---|---|---|
| Age (years) | 25 (22.5–28.5) | 25.0 (24.0–28.0) |
| Weight (kg) | 66.3 (66.3–72.5) | 69.1 (67.3–72.8) |
| Height (cm) | 177.0 (172.8–187.5) | 181.0 (178.5–185.0) |

BA: β-alanine group; PL: Placebo group. Data are presented as median (IQR).

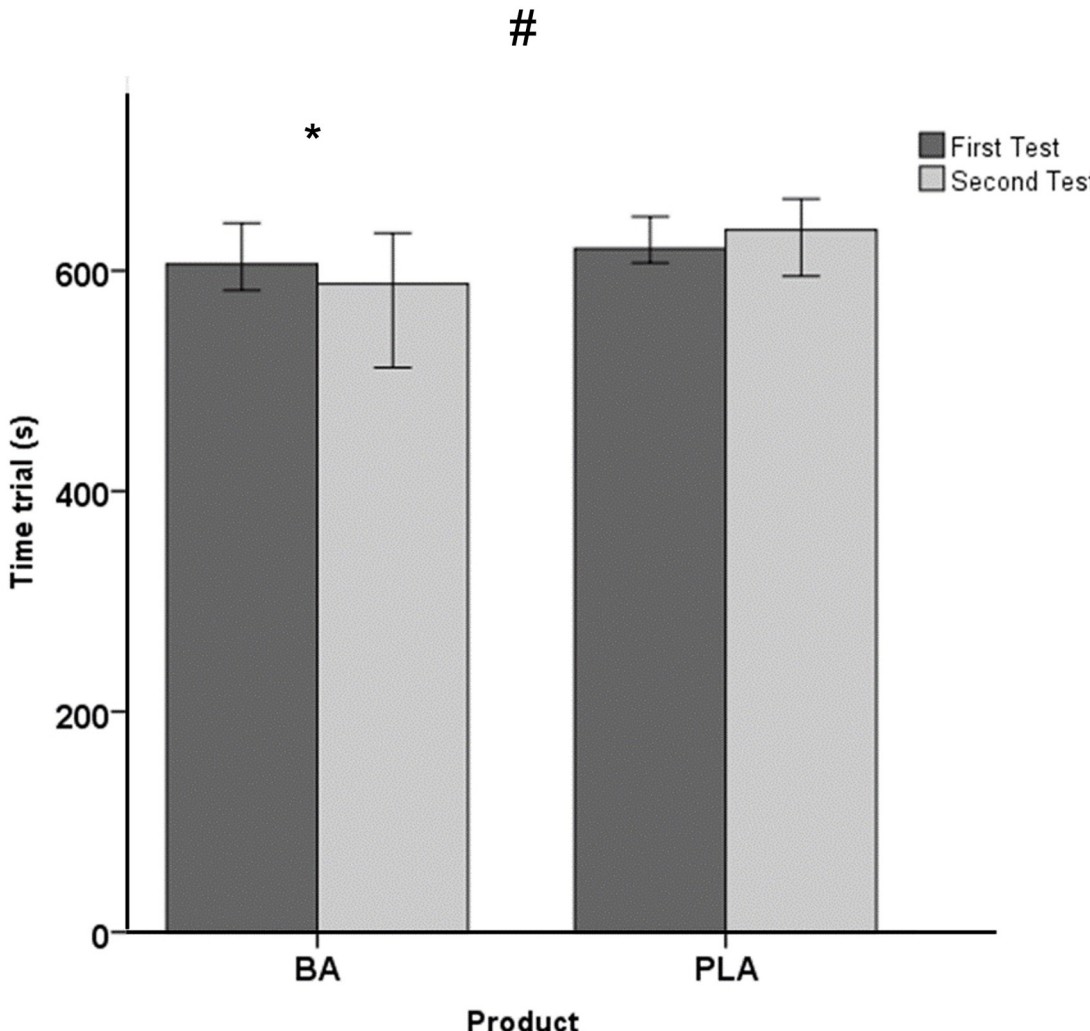

**Fig 2. Time to complete the time trial (median +/- 95% confidence interval) in both groups before and after supplementation (PLA: Placebo group; BA: β-Alanine group).** # $p < 0.05$ for the comparison in the evolution of the placebo and β-Alanine groups (ANOVA for repeated measures); * $p < 0.05$ for within-group comparisons in the groups (post hoc analysis, Bonferroni tests).

time to complete the time-trial. Moreover, a one-week high dose of β-alanine supplementation seems to minimise the fatigue that is associated with a cycling training camp. Ergogenic aid in periods of high load of training would be an interesting strategy.

Remarkably, the results from this study demonstrated a notable enhancement (approximately 5%) in the time taken to complete the 4.5 km uphill time trial following seven days of β-alanine supplementation, while the cyclists who ingested the placebo maintained their performance (i.e., the performance was no significantly impaired (~2%)). Our results are in accordance with a previous study [19] that analysed the effect of chronic β-alanine supplementation (four weeks and 6.4 gr per day) and found an improvement in 4 km flat time trial in trained cyclists. However, another study found an increase in muscle carnosine but no improvements in short-duration sprint performance throughout simulated road-race cycling or in a 4 km uphill time-trial performance that was conducted at the end of this cycling test by an amateur cyclist [20]. There are several reasons to explain these controversial results. First, it has been previously reported that exercise duration can determine the beneficial effect of the β-alanine

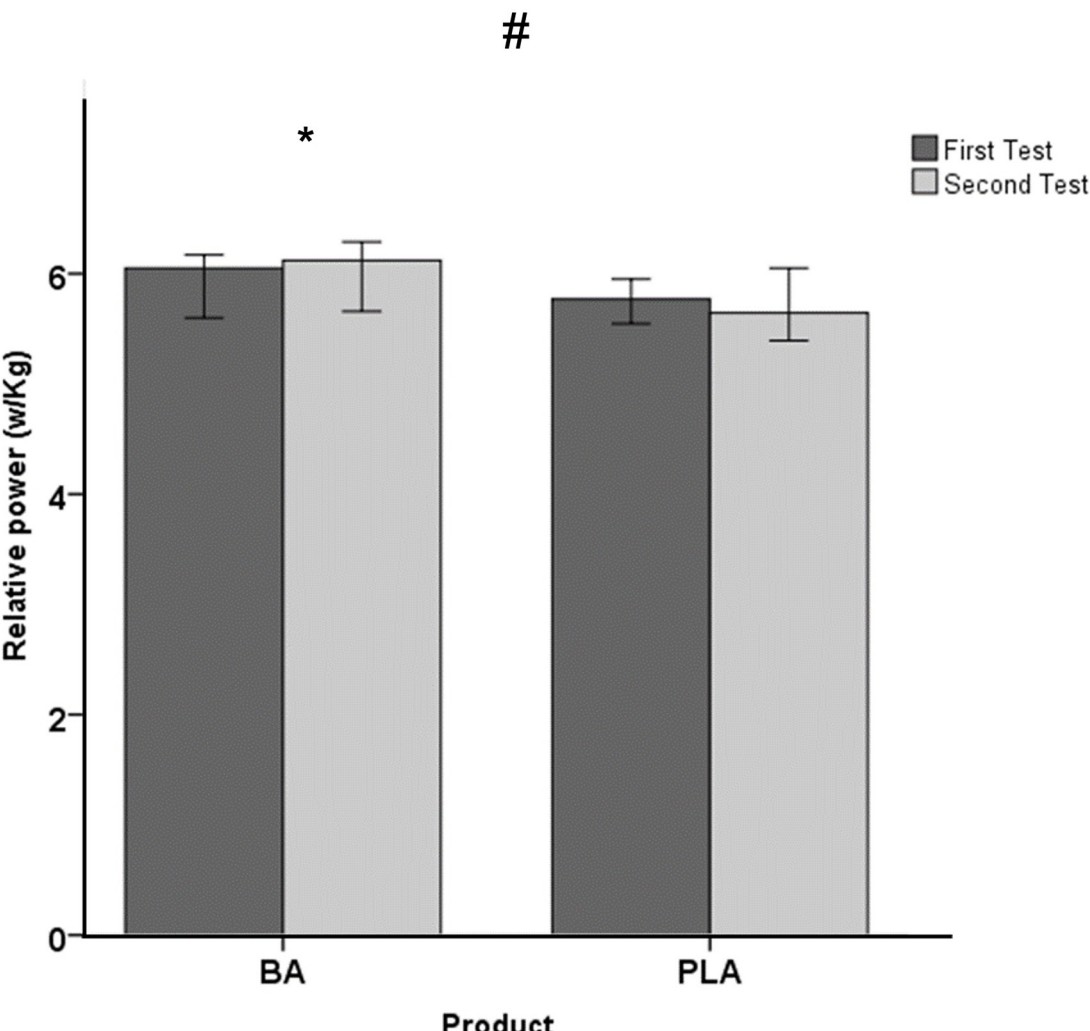

**Fig 3. Relative mean power (median +/- 95% confidence interval) during the time trial and in both groups before and after supplementation (PLA: Placebo group; BA: β-Alanine group).** # p < 0.05 for the comparison in the evolution of the placebo and β-Alanine groups (ANOVA for repeated measures); * p < 0.05 for within-group comparisons in the groups (post hoc analysis, Bonferroni tests).

supplementation. The duration of the uphill time-trial performed in the present research was around 10 minutes, which is in the range of efficacy reported in a previous meta-analysis (from 30 s to 10 min of duration) [14]. Second, the training status of the sample used in the study can also affect β-alanine supplementation because it is lower in nontrained individuals than in trained individuals [14]. In this sense, previous studies regarding cyclist and β-alanine [18–20] were developed using amateur or trained cyclists but not with a World Tour cyclist, like the present study. This fact can also be taken into consideration to explain our data. Finally, the supplementation protocol affects β-alanine effect and also the paresthesia symptoms. In the present study supplementation protocol took a duration of seven days with four daily intakes of 5 g of β-alanine (20 g/day; 155 g in total), whereas in the previously evidence chronic supplementation used a range between 1.5–6.4 g/day and from 4 to 10 weeks of duration [7,14]. The current study's supplement plan employed a slow-release formula, potentially speeding up the rise in carnosine levels in muscles within just a week. Additionally, reducing tingling sensations could be a helpful step before a major race or event.

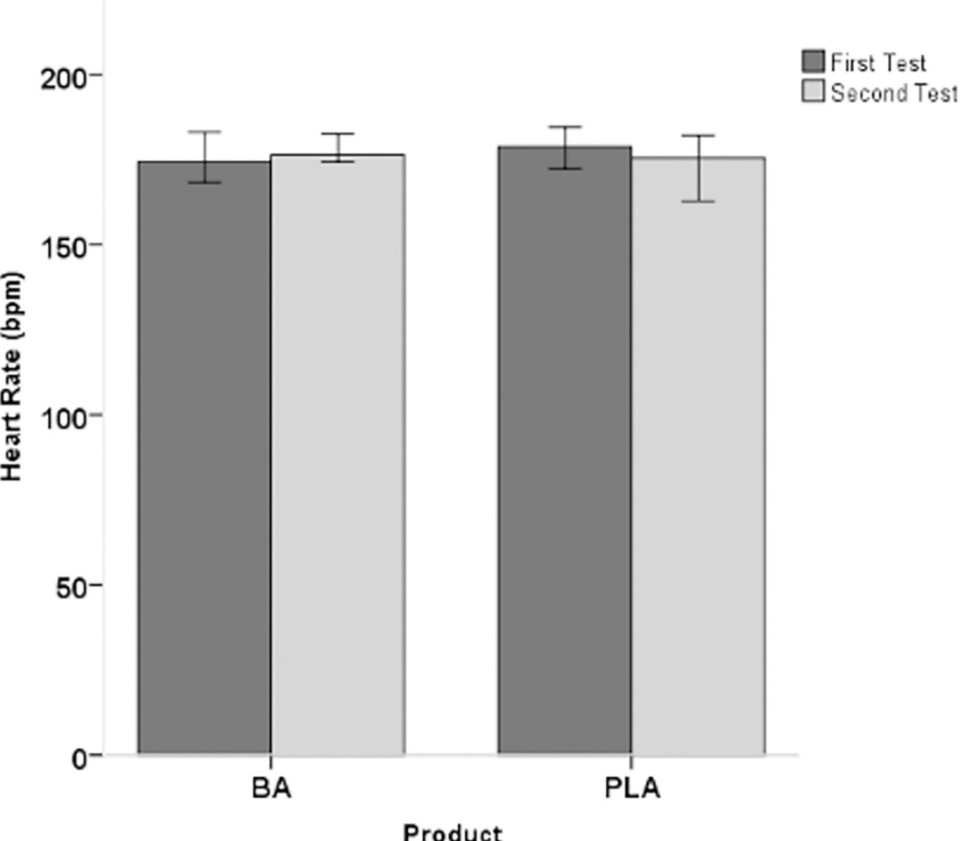

**Fig 4. Heart rate (median +/- 95% confidence interval) in both groups before and after supplementation (PLA: Placebo group; BA: β-Alanine group).**

Based on previous studies where participants ingested a totally of 134.4–448 gr of β-alanine [22,25–27], it is reasonable to assume that 155 gr β-alanine supplementation caused a significant increment of carnosine content in both muscle fibres (type I and II) promoting an increase in the intracellular buffer capacity of the muscle. These physiological effects associated with carnosine can explain the findings observed in the present study. First, muscle buffer capacity can produce a greater tolerance of sustained anaerobic activity [7], which is associated to efforts such as the 4.5 km uphill time trial that was performed by the cyclist in the present research. Therefore, this type of exercise increases reliance on glycolysis to maintain ATP supply, increasing in cellular production of $H^+$, which contributes to fatigue by muscle acidosis [4]. Our data suggest that this mechanism may be attenuated by the supplementation of β-alanine proposed. In addition, the greater uphill time-trial performance observed after β-alanine supplementation can be associated with the higher muscle force production by cross-bridge formation due to the higher $Ca^{+2}/H^+$ exchanger in the fibre, increasing $H^+$ binding to carnosine and producing $Ca^{+2}$ unloading at the sarcomere [9,10].

Interestingly, β-alanine reduced fatigue and improved performance during the training camp, while PLA tend to decrease uphill time-trial performance. During the training camps, the load used to increase and over-reaching is common [21]. However, our results suggest a lower fatigue and higher performance in BA during the uphill time trial test. This is related also with the blood lactate concentration observed during the test, where the participants of BA group showed a trend to increase anaerobic glycolysis and obtain higher blood lactate

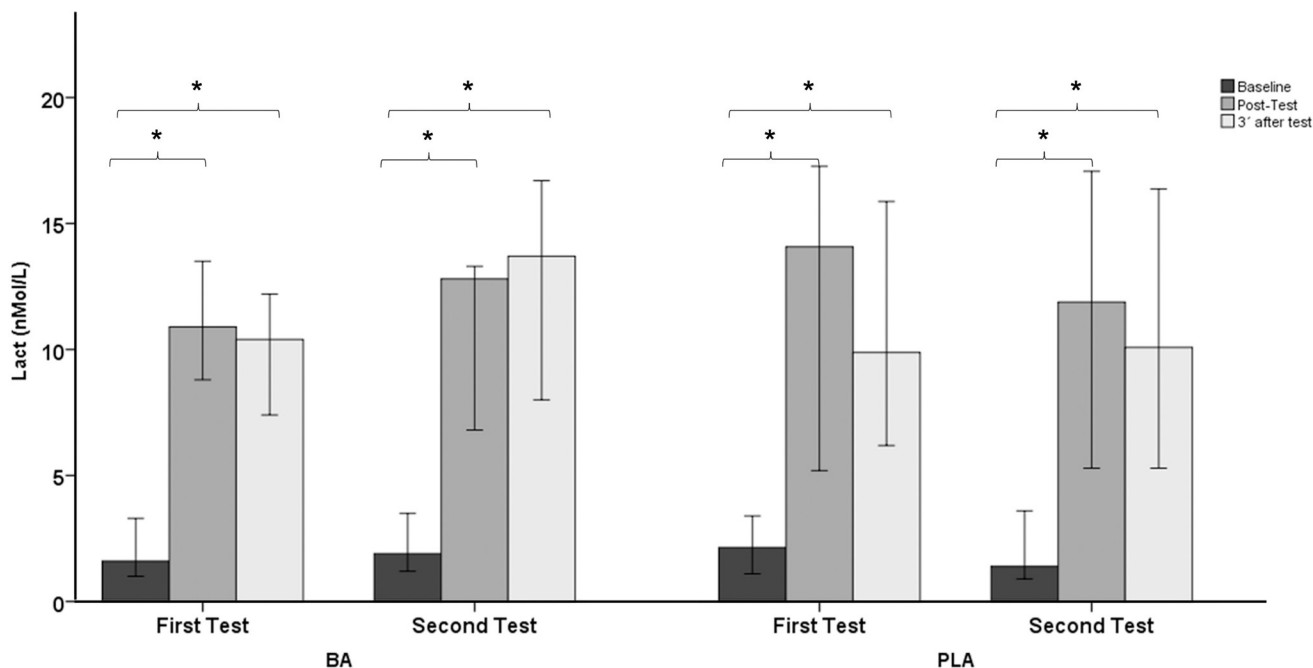

**Fig 5. Blood lactate concentration (median +/- 95% confidence interval) in both groups before and after supplementation (PLA: Placebo group; BA: β-Alanine group).**

concentration at the end of the test. Meanwhile, the PLA group showed the opposite effect, reporting lower values of blood lactate concentration. This finding may be related to the anti-oxidant, and antiglycating and ion-chelating effect of carnosine [11–13]. Although this theory

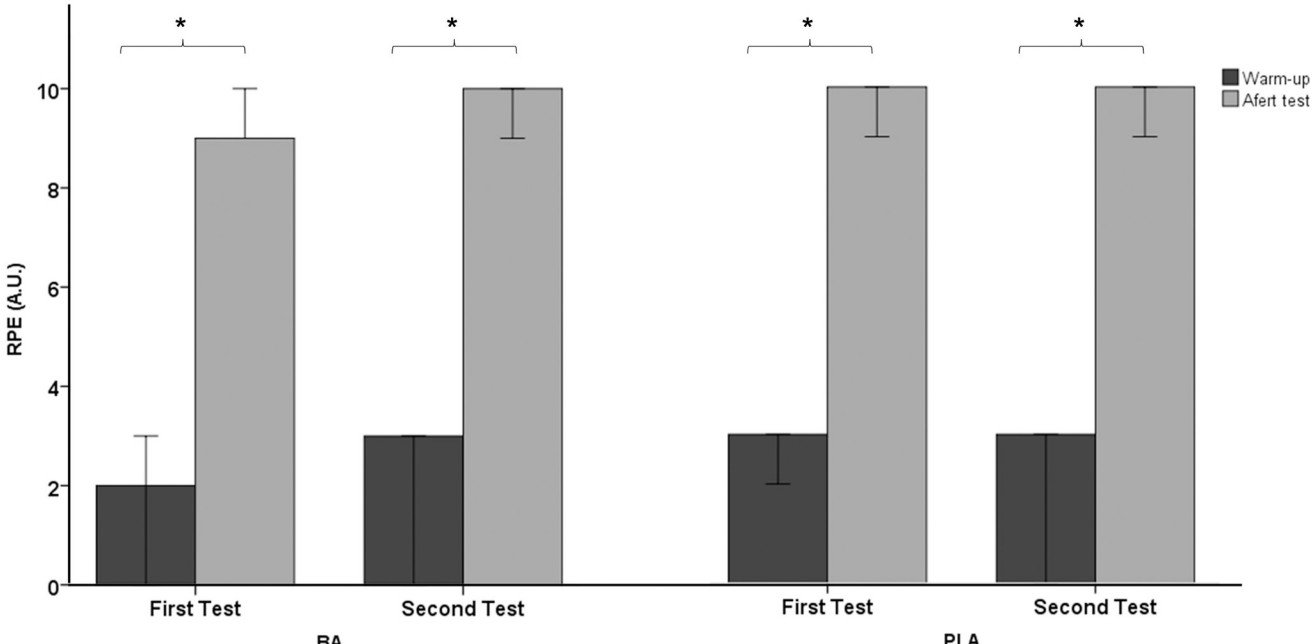

**Fig 6. Rating of Perceived Exertion (median +/- 95% confidence interval) in both groups before and after supplementation (PLA: Placebo group; BA: β-Alanine group).** * $p < 0.05$ for within-group comparisons in the groups (post hoc analysis, Bonferroni tests).

**Table 2. Training load characteristics during the five days of intense training.**

| Outcome | PL (n = 6) | BA (n = 5) | *p* value |
|---|---|---|---|
| Volume (hours) | 4.92 (4.80–5.08) | 5.10 (4.96–5.14) | 0.177 |
| Distance (km) | 154.82 (149.80–158.35) | 159.45 (153.53–160.34) | 0.329 |
| Mean Power (w) | 178.67 (173.31–195.10) | 187.50 (179.66–201.67) | 0.429 |
| Normalized Power (w) | 258.90 (236.40–277.75) | 265.00 (246.70–270.60) | 0.931 |
| Training Stress Score | 258.78 (225.01–323.02) | 276.60 (255.89–388.24) | 0.429 |

BA: β-alanine group; PL: Placebo group. Data are presented as median (interquartile range).

is not confirmed in humans yet, evidence suggests the possible beneficial effect of β-alanine supplementation in the recovery process [7]. This finding could be particularly valuable in training camps, especially when cyclists are undergoing intense training phases with heavy loads or pushing themselves to their limits.

However, it's important to note some limitations that should be considered when interpreting the data. First, muscle carnosine content was not measured in the present study (via biopsy or magnetic resonance spectrometry). This outcome has not been evaluated because an invasive technique that can condition the training process of a professional cyclist is necessary to obtain it. Therefore, it was not considered relevant by the team's medical services and the researchers. Moreover, another limitation was the small number of participants who took part in the research study, due to the low availability of these elite professional cyclists. It is very difficult to get access to this type of athlete and this would be considered as one of the strengths of the present study. In addition, another of the strengths is the practical application of the results obtained in the present study to the real field. Therefore, from a practical application point of view, if coaches or cyclists want to obtain a beneficial performance effect in an uphill time trial (4.5 km), or they want to reduce the fatigue in a training camp, then they may include a sustained-release β-alanine supplementation during seven days with four daily intakes of 5 g of β-alanine (20 g/day).

## Conclusion

The data in the present study demonstrated the effectiveness of one-week high dose of β-alanine during a cycling training camp in World Tour cyclists on uphill time-trial performance

## Supporting information

**S1 Checklist. CONSORT 2010 checklist of information to include when reporting a randomised trial\*.**
(DOC)

**S1 File.**
(XLSX)

**S2 File.**
(PDF)

## Acknowledgments

The authors thank the staff of the Department of Sports Physiology of the Catholic University of Murcia (UCAM) for their technical support. We also thanks Martinez Nieto S.A. for manufacturing and providing the product and placebo used in this study.

## Author Contributions

**Conceptualization:** Domingo Jesús Ramos-Campo, Francisco Javier López-Román.

**Data curation:** Francisco Javier López-Román, Raquel Ortolano, Antonio Jesús Luque-Rubia, Natalia Ibáñez-Soroa, Vicente Ávila-Gandía.

**Formal analysis:** Silvia Pérez-Piñero, Francisco Javier López-Román, Raquel Ortolano.

**Investigation:** Silvia Pérez-Piñero, Antonio Torregrosa-García, Vicente Ávila-Gandía.

**Methodology:** Silvia Pérez-Piñero, Domingo Jesús Ramos-Campo, Antonio Torregrosa-García, Vicente Ávila-Gandía.

**Supervision:** Silvia Pérez-Piñero, Francisco Javier López-Román, Antonio Jesús Luque-Rubia, Luis Andreu-Caravaca.

**Writing – original draft:** Francisco Javier López-Román.

**Writing – review & editing:** Francisco Javier López-Román, Antonio Jesús Luque-Rubia, Luis Andreu-Caravaca, Vicente Ávila-Gandía.

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
