## [Decision Letter · Decision Letter 0]

2 Apr 2024

PONE-D-24-04397Effect of high-dose β-Alanine supplementation on uphill cycling performance in World Tour cyclists: A randomised controlled trial.PLOS ONE

Dear Dr. López-Román,

Thank you for submitting your manuscript to PLOS ONE. After careful consideration, we feel that it has merit but does not fully meet PLOS ONE’s publication criteria as it currently stands. Therefore, we invite you to submit a revised version of the manuscript that addresses the points raised during the review process.

We look forward to receiving your revised manuscript.

Kind regards,

Fenghua Sun

Academic Editor

PLOS ONE

2. We note that you have selected “Clinical Trial” as your article type. PLOS ONE requires that all clinical trials are registered in an appropriate registry (the WHO list of approved registries is at https://www.who.int/clinical-trials-registry-platform/network/primary-registries" https://www.who.int/clinical-trials-registry-platform/network/primary-registries and more information on trial registration is at http://www.icmje.org/about-icmje/faqs/clinical-trials-registration/). Please state the name of the registry and the registration number (e.g. ISRCTN or ClinicalTrials.gov) in the submission data and on the title page of your manuscript. a) Please provide the complete date range for participant recruitment and follow-up in the methods section of your manuscript. b) If you have not yet registered your trial in an appropriate registry, we now require you to do so and will need confirmation of the trial registry number before we can pass your paper to the next stage of review. Please include in the Methods section of your paper your reasons for not registering this study before enrolment of participants started. Please confirm that all related trials are registered by stating: “The authors confirm that all ongoing and related trials for this drug/intervention are registered”. Please see http://journals.plos.org/plosone/s/submission-guidelines#loc-clinical-trials for our policies on clinical trials.

3. In the online submission form you indicate that your data is not available for proprietary reasons and have provided a contact point for accessing this data. Please note that your current contact point is a co-author on this manuscript. According to our Data Policy, the contact point must not be an author on the manuscript and must be an institutional contact, ideally not an individual. Please revise your data statement to a non-author institutional point of contact, such as a data access or ethics committee, and send this to us via return email. Please also include contact information for the third party organization, and please include the full citation of where the data can be found.

4. We note that the original protocol that you have uploaded as a Supporting Information file contains an institutional logo. As this logo is likely copyrighted, we ask that you please remove it from this file and upload an updated version upon resubmission.

Reviewers' comments:

Reviewer's Responses to Questions

**Comments to the Author**

1. Is the manuscript technically sound, and do the data support the conclusions?

Reviewer #1: No

Reviewer #2: Yes

Reviewer #3: Partly

2. Has the statistical analysis been performed appropriately and rigorously? 

Reviewer #1: No

Reviewer #2: Yes

Reviewer #3: N/A

3. Have the authors made all data underlying the findings in their manuscript fully available?

Reviewer #1: Yes

Reviewer #2: Yes

Reviewer #3: Yes

4. Is the manuscript presented in an intelligible fashion and written in standard English?

Reviewer #1: Yes

Reviewer #2: Yes

Reviewer #3: Yes

5. Review Comments to the Author

Reviewer #1: A two-arm randomized controlled trial was conducted which aimed to investigate the effectiveness of ergogenic aids (beta-alanine) on uphill cycling performance (n=11). The conclusions are unclear.

Major revision:

A comprehensive reanalysis is required.

1- Since sample sizes of 5 and 6 are too small to be tested for a normal distribution, use nonparametric tests for the analysis, i.e., use the Wilcoxon rank sum test instead of the t-test. Furthermore, summarize data using medians and first and third quartiles instead of means and standard deviations.

2- Line 214: In the statistical methods section, list and describe the use of all statistical methods. Due to the small sample size, nonparametric statistical methods are needed.

Minor revisions:

1- Line 132: Indicate the statistical testing method which attains 80% power.

2- The standard statistical term for average is mean.

3- Table 1: In the table header, indicate the sample sizes.

4- Include a table of subject characteristics (age, race, BMI, etc.) by randomization arm.

Reviewer #2: This study included a small cohort of elite cyclists randomized to placebo or high-dose B-alanine supplementation for 7 days. This was a well-controlled and presented study. However, there are several concerns that should be addressed and revised.

Introduction, line 63-64, this is not entirely true. The buffering capacity enhances aerobic byproduct H+ accumulation as well. Although anaerobic systems initiate the production of the byproducts, beyond 90-120 seconds, the shift to aerobic metabolism is ramped up. B-alanine has been reported to benefit both types of activities, arguably more beneficial in endurance athletes. This is then expanded on somewhat in lines 73-79 but comes up short-sighted when referencing only meta analyses and not primary literature of the actual effect. Please revise. Additionally, suggesting the anaerobic impact of B-alanine as more pronounced counters the proposed purpose and hypothesis in the present study that evaluated an aerobic task.

Introduction, lines 84-85: The ceiling effect mentioned here is from the training/performance perspective rather than the effect of B-alanine. If any individual consumes B-alanine, they will respond differently depending on if their carnosine stores are not optimized. Loading B-alanine can enhance their carnosine, if it is lower than optimal. Loaded or optimized carnosine can then, in turn, shift the training adaptations but those adaptations would be less for trained individuals because they are close to the top of peak performance whereas the untrained individual has the much more growth potential to get to peak.

Line 95: please be consistent with use of B-ala vs B-alanine, completely spelled out is presented throughout the majority of this manuscript. Also: Line 142 spells out “beta” and does not use the B symbol.

Figures 1 and 2 seem to both be the CONSORT flow diagram and they do not share the same sample values throughout the flow. Please include the referred-to training camp scheme and ensure the CONSORT flow is accurate. The CONSORT flow diagram is currently presented in the results section and should be placed in the methods, referenced in the participants section.

Do the authors think the participants in the B-alanine group experienced greater increases in intramuscular carnosine (although not measured) because they consumed B-alanine, L-histidine, and carnosine in the daily product doses?

Were these athletes free of caffeine and other products that may confine the interpretation of the results.

Did the authors document other supplement use? Were the athletes all B-alanine naive (they have never used BA)?

Please mention the type of bike the cyclists were assessed on and trained on. Was it their own personal bike on a trainer? Was each cyclist supplied the Garmin Vector 3 pedals for their bikes? It is not clear in the methods.

Results: please present the power of each analysis conducted. Although the pre-determined sample size was (mostly) met, did the power of each analysis meet desired level?

Please revise the figures to a higher quality. The axes are very difficult to read. Additionally, please present the data as mean +/- SE of the data, not an arbitrary +/- 1SE.

Please present the post-hoc t-tests with additional context of confidence intervals for each comparison. The response (broadly using the data provided) is not as pronounced as the figures indicate because the y-axis is zoomed-in so far.

Results: please refrain from the use of “trend” with such a small sample size per group this “trend” could be null if the sample were doubled. It could also be more pronounced but without the additional data, it is either significant or not, it cannot trend.

Line 289: the dosing described in the methods included this description of 5 g, 4 times a day for 7 days. This would mathematically result in 140g total for the week. However, the dose was listed as 31 total doses of 5 grams. Please revise here or in the methods (or both) to ensure consistent messaging of the dosing.

What is also not clear, and can be clarified with a figure, is the timeline of events. Pre-testing was considered day 1 and the first dose was consumed then, they trained for 5 days and day 7 was post-testing. With this condensed 7-day cycle, the dosing would be less than 140g (5g/dose, 4x per day) or was there an additional day in there somewhere to hit 155g BA?

The results indicate a significant improvement in power and time trial performance for 4.5km. Their average was around 10 minutes to completion. How do these athletes compare to others with similar ability? They are described as World Tour cyclists, but do they compete in Olympic games, Tour de France, 10 minutes for a 4.5 km ride is “slow” for 20-30 year old “average” cyclists nationally (see Gough et al 2021). It is understood this is at 5%grade, but nearly 2-3 minutes longer than other studies report for this 4.5 km distance.

Lines 312-316: the authors reported no significant effect of lactate yet this section of discussion is reaching to suggest lactate contributed to performance enhancement.

There are some places throughout the manuscript that need revised writing for grammatical clarity (ex. Lines 292-294 and others (lines 271, 324, 326)).

Reviewer #3: Overall, the article is written in an orderly and linear manner, and the concepts are clearly expressed.

The understandability and overall quality of the manuscript can certainly be improved by addressing several issues that I raise here.

In line 37, writing just "to improve cycling performance" may seem a bit excessive. It would be beneficial to also highlight the potential role of β-alanine in improving recovery power. This aspect is particularly relevant in the context of a training camp, where fatigue levels can increase alongside training intensity. It is reasonable to consider that reducing chronic fatigue, possibly aided by β-alanine supplementation, could lead to overall higher performances in both groups under similar conditions. This aligns with the assertion made in line 67 of the study.

L54-57 – it is useful to insert a transition sentence between the concept of training and that of integration (ergogenic aids probably do not fit at all with "training strategies")

In line 191, there appears to be a significant limitation in quantifying performance (and load) in this sentence. It's important to provide details about the protocol used for the incremental test and the duration of the test steps. Equating Functional Threshold Power (FTP), defined as the sustainable power for 1 hour, with the power at 4 mmol/l, especially calculated on short steps, may be too approximate. Clarifying this point would enhance the rigor and accuracy of the analysis

In line 222, it's essential to consider whether there were any significant changes in body mass throughout the study. Additionally, it may be worth exploring whether there is a potential effect of β-alanine intake on body mass or body composition, especially considering that power changes less than relative power. This additional analysis could provide valuable insights into the broader effects of β-alanine supplementation beyond performance outcomes.

6. PLOS authors have the option to publish the peer review history of their article (what does this mean?). If published, this will include your full peer review and any attached files.

Reviewer #1: No

Reviewer #2: No

Reviewer #3: No

---

## [Author Response · Author response to Decision Letter 0]

27 Apr 2024

AUTHORS REPLY

A: First of all, we would like to express our gratitude to Reviewer 1 for providing us with comments and suggestions to help improve the quality of our paper. We have answered your comments point-by-point below and have marked our changes in red in the new version of the manuscript.

Reviewer #1: A two-arm randomized controlled trial was conducted which aimed to investigate the effectiveness of ergogenic aids (beta-alanine) on uphill cycling performance (n=11). The conclusions are unclear.

Major revision:

A comprehensive reanalysis is required.

1- Since sample sizes of 5 and 6 are too small to be tested for a normal distribution, use nonparametric tests for the analysis, i.e., use the Wilcoxon rank sum test instead of the t-test. Furthermore, summarize data using medians and first and third quartiles instead of means and standard deviations.

2- Line 214: In the statistical methods section, list and describe the use of all statistical methods. Due to the small sample size, nonparametric statistical methods are needed.

ANSWER: We thank the reviewer for the valuable review. Below we reply to points 1 and 2: 

Non-parametric ANOVA is an alternative method to the traditional Analysis of Variance widely used in statistical analysis. This method is employed when the assumptions of normality and homogeneity of variance are not met. Non-parametric ANOVA is utilized in cases where the data distribution is unknown or the sample size is small. The method relies on data rankings, rendering it more robust and less sensitive to outliers. 

Traditional analysis of variance is a statistical technique commonly used for analyzing data with multiple groups. 

Some key points to understand traditional ANOVA are: 

• Normality assumption: Traditional ANOVA assumes that the data follow a normal distribution. This assumption is necessary because the test is based on calculating the means and variances of the data. 

• Equal variances assumption: Traditional ANOVA assumes that the variances of the groups are equal. This assumption is necessary because the test uses the relationship between the between-group variance and the within-group variance to determine if there is a significant difference in means. 

• Sensitivity to outliers: Traditional ANOVA is sensitive to outliers, which can skew the means and variances of the data. Outliers can lead to incorrect conclusions about the significance of mean differences. 

• Post hoc tests are necessary: If the test finds a significant difference in means, post hoc tests are necessary to determine which groups are significantly different from each other. 

Indeed, our samples have a small size: 5 subjects in the experimental group and 6 subjects in the control group. 

Therefore, as indicated in lines 196-8, we conducted tests for normality (Shapiro-Wilk test) and homoscedasticity (Levene's Test) to ascertain if we could apply parametric tests to our analysis. All study variables were normally distributed, and the sample variances were equal, so it was decided to apply parametric tests as indicated in lines 198-9: repeated measures ANOVA with two factors under study (an intra-subject factor (test) and an inter-subject factor (product)). In the event of a significant ANOVA result, a Bonferroni test was conducted (lines 199-200). 

Minor revisions:

1- Line 132: Indicate the statistical testing method which attains 80% power.

ANSWER: The test has been added. 

2- The standard statistical term for average is mean.

ANSWER: Changed

3- Table 1: In the table header, indicate the sample sizes.

ANSWER: Added. 

4- Include a table of subject characteristics (age, race, BMI, etc.) by randomization arm.

ANSWER: A table with the descriptive data of the sample has been added.

 

A: First of all, we would like to express our gratitude to Reviewer 2 for providing us with comments and suggestions to help improve the quality of our paper. We have answered your comments point-by-point below and have marked our changes in red in the new version of the manuscript.

Reviewer #2: This study included a small cohort of elite cyclists randomized to placebo or high-dose B-alanine supplementation for 7 days. This was a well-controlled and presented study. However, there are several concerns that should be addressed and revised.

Introduction, line 63-64, this is not entirely true. The buffering capacity enhances aerobic byproduct H+ accumulation as well. Although anaerobic systems initiate the production of the byproducts, beyond 90-120 seconds, the shift to aerobic metabolism is ramped up. B-alanine has been reported to benefit both types of activities, arguably more beneficial in endurance athletes. This is then expanded on somewhat in lines 73-79 but comes up short-sighted when referencing only meta analyses and not primary literature of the actual effect. Please revise. Additionally, suggesting the anaerobic impact of B-alanine as more pronounced counters the proposed purpose and hypothesis in the present study that evaluated an aerobic task.

ANSWER: We thank the reviewer for the comment. We would like to clarify that the task performed by cyclists is not aerobic, but mixed and is limited by the energy pool obtained by anaerobic metabolism. During aerobic processes the H+ generated are resynthesised in the mitochondria.

Hargreaves, M., Spriet, LL Skeletal muscle energy metabolism during exercise. Nat Metab 2 , 817–828 (2020). https://doi.org/10.1038/s42255-020-0251-4

Introduction, lines 84-85: The ceiling effect mentioned here is from the training/performance perspective rather than the effect of B-alanine. If any individual consumes B-alanine, they will respond differently depending on if their carnosine stores are not optimized. Loading B-alanine can enhance their carnosine, if it is lower than optimal. Loaded or optimized carnosine can then, in turn, shift the training adaptations but those adaptations would be less for trained individuals because they are close to the top of peak performance whereas the untrained individual has the much more growth potential to get to peak.

ANSWER: We fully agree with the reviewer's comment.

Line 95: please be consistent with use of B-ala vs B-alanine, completely spelled out is presented throughout the majority of this manuscript. Also: Line 142 spells out “beta” and does not use the B symbol.

ANSWER: The term has been unified throughout the manuscript. 

Figures 1 and 2 seem to both be the CONSORT flow diagram and they do not share the same sample values throughout the flow. Please include the referred-to training camp scheme and ensure the CONSORT flow is accurate. The CONSORT flow diagram is currently presented in the results section and should be placed in the methods, referenced in the participants section.

ANSWER: This is a mistake. Figure 2 has been deleted. In addition, it has been included in the Method.

Do the authors think the participants in the B-alanine group experienced greater increases in intramuscular carnosine (although not measured) because they consumed B-alanine, L-histidine, and carnosine in the daily product doses?

ANSWER: We do believe so, as this is the conclusion of scientific papers that have measured it. Without measuring it, however, we cannot say for sure. However, taking the reviewer's comment, we will look at intramuscular carnosine in future studies.

Were these athletes free of caffeine and other products that may confine the interpretation of the results.

ANSWER: Yes. This was one of the exclusion criteria for the study: 

“iv) ingested β-alanine supplementation in the two months before the study; and v) used other performance enhancer supplements during the study or 15 days before it”.

Did the authors document other supplement use? Were the athletes all B-alanine naive (they have never used BA)?

ANSWER: Inclusion and exclusion criteria were checked. Athletes had used beta alanine at lower doses during the previous cycling seasons. The inclusion and exclusion criteria were set so that previous use would not have an effect on our study. In addition, it is an odourless and tasteless product, and the placebo was of similar presentation. They were asked if they thought they had taken the product: 

“From the eleven subjects, eight believed they consumed the experimental product (four from the BA group, and four from the PLA group), while one believed be-longing to the control product (actually in the PLA group) and two doubt (one from BA and one from PLA)”.

Please mention the type of bike the cyclists were assessed on and trained on. Was it their own personal bike on a trainer? Was each cyclist supplied the Garmin Vector 3 pedals for their bikes? It is not clear in the methods.

ANSWER: All the cyclists used the same model of bicycle, which is the one provided by the team for the season. All the bicycles were adapted to each athlete with biomechanical studies. Yes, the pedals were provided and calibrated. This information has been added in the method.

Results: please present the power of each analysis conducted. Although the pre-determined sample size was (mostly) met, did the power of each analysis meet desired level?

ANSWER: Power values have been added in the text. 

Please revise the figures to a higher quality. The axes are very difficult to read. Additionally, please present the data as mean +/- SE of the data, not an arbitrary +/- 1SE.

ANSWER: The figures have been modified for higher quality. Regarding the SE, this was a typing error which has now been corrected.

Please present the post-hoc t-tests with additional context of confidence intervals for each comparison. The response (broadly using the data provided) is not as pronounced as the figures indicate because the y-axis is zoomed-in so far.

ANSWER: Following the reviewer's recommendations, the Y-axis of the figures have been modified. In addition, confidence intervals have been added in the text.

Results: please refrain from the use of “trend” with such a small sample size per group this “trend” could be null if the sample were doubled. It could also be more pronounced but without the additional data, it is either significant or not, it cannot trend.

ANSWER: Thank you for your comment. It was amended accordingly. 

Line 289: the dosing described in the methods included this description of 5 g, 4 times a day for 7 days. This would mathematically result in 140g total for the week. However, the dose was listed as 31 total doses of 5 grams. Please revise here or in the methods (or both) to ensure consistent messaging of the dosing.

What is also not clear, and can be clarified with a figure, is the timeline of events. Pre-testing was considered day 1 and the first dose was consumed then, they trained for 5 days and day 7 was post-testing. With this condensed 7-day cycle, the dosing would be less than 140g (5g/dose, 4x per day) or was there an additional day in there somewhere to hit 155g BA?

ANSWER: The subjects commenced supplementation after the physical tests (immediately following the snack) and consumed the final intake after breakfast on the day of the final test (a total of 7 days and 3 additional doses, 31 servings in total).

The results indicate a significant improvement in power and time trial performance for 4.5km. Their average was around 10 minutes to completion. How do these athletes compare to others with similar ability? They are described as World Tour cyclists, but do they compete in Olympic games, Tour de France, 10 minutes for a 4.5 km ride is “slow” for 20-30 year old “average” cyclists nationally (see Gough et al 2021). It is understood this is at 5%grade, but nearly 2-3 minutes longer than other studies report for this 4.5 km distance.

ANSWER: We believe that the results obtained by the cyclists are excellent since the time difference between a flat time trial (TT) and an uphill time trial is very significant. Antón et al. (2007), with highly trained individuals, observed that the time elapsed during the uphill time trial (2.9-5.2%) of 6.7 km and the flat time trial (FTT) of 14 km was 1115 ± 59 s (18.6 ± 1.0 min; 21.6 ± 1.2 km/h), and 1185 ± 35 s (19.8 ± 0.6 min; 42.5 ± 1.3 km/h), respectively.

Antón MM, Izquierdo M, Ibáñez J, Asiain X, Mendiguchía J, Gorostiaga EM. Flat and uphill climb time trial performance prediction in elite amateur cyclists. Int J Sports Med. 2007 Apr;28(4):306-13. doi: 10.1055/s-2006-924356. Epub 2006 Oct 6. PMID: 17024629.

Furthermore, in our study, the athletes were world-class, competing in the Tour de France. These athletes exhibited average relative powers of around 6 W/kg. Sanders et al. describe the workload and intensity of races in a Grand Tour, where in time trials, they achieve average relative power of 5.14 W/kg.

Sanders, D.; Heijboer, M. Physical demands and power profile of different stage types within a cycling grand tour. https://doi.org/10.1080/17461391.2018.1554706 2018, 19, 736–744, doi:10.1080/17461391.2018.1554706

Lines 312-316: the authors reported no significant effect of lactate yet this section of discussion is reaching to suggest lactate contributed to performance enhancement.

ANSWER: The discussion refers to the trend, as we did not find significant results, possibly because of the small n.

There are some places throughout the manuscript that need revised writing for grammatical clarity (ex. Lines 292-294 and others (lines 271, 324, 326)).

ANSWER: The wording of the parts pointed out by the reviewer has been revised and modified to improve the reader's understanding.

 

A: First of all, we would like to express our gratitude to Reviewer 3 for providing us with comments and suggestions to help improve the quality of our paper. We have answered your comments point-by-point below and have marked our changes in red in the new version of the manuscript.

Reviewer #3: Overall, the article is written in an orderly and linear manner, and the concepts are clearly expressed.

The understandability and overall quality of the manuscript can certainly be improved by addressing several issues that I raise here.

In line 37, writing just "to improve cycling performance" may seem a bit excessive. It would be beneficial to also highlight the potential role of β-alanine in improving recovery power. This aspect is particularly relevant in the context of a training camp, where fatigue levels can increase alongside training intensity. It is reasonable to consider that reducing chronic fatigue, possibly aided by β-alanine supplementation, could lead to overall higher performances in both groups under similar conditions. This aligns with the assertion made in line 67 of the study.

ANSWER: Information on the reviewer's comments has been added.

L54-57 – it is useful to insert a transition sentence between the concept of training and that of integration (ergogenic aids probably do not fit at all with "training strategies")

ANSWER: We have modified the paragraph according to the reviewer’s comment: Developing the most effective and efficient training method to optimize cycling performance has been the focus of many scientists and coaches. While training strategies are crucial, it's also important to consider the integration of legal ergogenic aids, which is a popular approach among athletes, particularly cyclists, seeking to maximize their physical performance

In line 191, there appears to be a significant limitation in quantifying performance (and load) in this sentence. It's important to provide details about the protocol used for the incremental test and the duration of the test steps. Equating Functional Threshold Power (FTP), defined as the sustainable power for 1 hour, with the power at 4 mmol/l, especially calculated on short steps, may be too approximate. Clarifying this point would enhance the rigor and accuracy of the analysis

ANSWER: We concur with the reviewer's comment. Previous studies have indicated that trained cyclists can sustain 100% of LT4.0 for 60 minutes, and FTP is considered equivalent to the power output at LT4 (Gavin et al., 2012). The test employed in our study was an incremental cycling test. Therefore, we utilized the methodology outlined in the article. Additionally, we have incorporated the following details in the text: The test commenced with an initial resistance of 110 W, with subsequent increments of 3

---

## [Decision Letter · Decision Letter 1]

21 May 2024

PONE-D-24-04397R1Effect of high-dose β-Alanine supplementation on uphill cycling performance in World Tour cyclists: A randomised controlled trial.PLOS ONE

Dear Dr. López-Román,

Thank you for submitting your manuscript to PLOS ONE. After careful consideration, we feel that it has merit but does not fully meet PLOS ONE’s publication criteria as it currently stands. Therefore, we invite you to submit a revised version of the manuscript that addresses the points raised during the review process.

**Although the reviewers appreciated your revisions, one concern about the statistical method has been raised. Please carefully consider this comment and make some further corrections, if possible. Thanks.**

We look forward to receiving your revised manuscript.

Kind regards,

Fenghua Sun

Academic Editor

PLOS ONE

Reviewers' comments:

Reviewer's Responses to Questions

**Comments to the Author**

1. If the authors have adequately addressed your comments raised in a previous round of review and you feel that this manuscript is now acceptable for publication, you may indicate that here to bypass the “Comments to the Author” section, enter your conflict of interest statement in the “Confidential to Editor” section, and submit your "Accept" recommendation.

Reviewer #1: (No Response)

Reviewer #2: (No Response)

Reviewer #3: All comments have been addressed

2. Is the manuscript technically sound, and do the data support the conclusions?

Reviewer #1: No

Reviewer #2: Yes

Reviewer #3: Yes

3. Has the statistical analysis been performed appropriately and rigorously? 

Reviewer #1: No

Reviewer #2: Yes

Reviewer #3: Yes

4. Have the authors made all data underlying the findings in their manuscript fully available?

Reviewer #1: Yes

Reviewer #2: (No Response)

Reviewer #3: Yes

5. Is the manuscript presented in an intelligible fashion and written in standard English?

Reviewer #1: Yes

Reviewer #2: Yes

Reviewer #3: Yes

6. Review Comments to the Author

Reviewer #1: Major revision:

My prior comment concerning using the appropriate statistical testing method based on the distribution of the data has not been adequately addressed. The sample sizes are too small to test for normal distributions; therefore, nonparametric tests are called for.

Prior Comment: A comprehensive reanalysis is required.

Since sample sizes of 5 and 6 are too small to be tested for a normal distribution, use nonparametric tests for the analysis, i.e., use the Wilcoxon rank sum test instead of the t-test. Furthermore, summarize data using medians and first and third quartiles instead of means and standard deviations.

Minor revision:

Line 101: For the power justification, indicate if the t-test was the test used for the"comparisons of means test".

Reviewer #2: The text calls out Figure 1 to represent study flow but the attached figure 1 is the CONSORT Diagram.

Please provide the manufacturer information for the bikes used.

The figures are still very blurry. The axes and legends cannot be read.

All other comments have been sufficiently addressed. Some responses to the reviewers should be reflected in the manuscript text and not just in the responses to reviewer comment document.

Reviewer #3: Thank you to the authors for the answers. The protocol used is correct, and the overall explanation of the findings is clear. However, I suggest checking these two articles for further improvement of the FTP concept.

Wong S, Burnley M, Mauger A, Fenghua S, Hopker J. Functional threshold power is not a valid marker of the maximal metabolic steady state. J Sports Sci. 2022 Dec;40(23):2578-2584. doi: 10.1080/02640414.2023.2176045. Epub 2023 Feb 20. PMID: 36803419.

Vinetti G, Rossi H, Bruseghini P, Corti M, Ferretti G, Piva S, Taboni A, Fagoni N. Functional Threshold Power Field Test Exceeds Laboratory Performance in Junior Road Cyclists. J Strength Cond Res. 2023 Sep 1;37(9):1815-1820. doi: 10.1519/JSC.0000000000004471. Epub 2023 Feb 2. PMID: 36692223; PMCID: PMC10448799.

7. PLOS authors have the option to publish the peer review history of their article (what does this mean?). If published, this will include your full peer review and any attached files.

Reviewer #1: No

Reviewer #2: No

Reviewer #3: **Yes: **Roberto Codella

---

## [Author Response · Author response to Decision Letter 1]

2 Jul 2024

AUTHORS REPLY

A: First of all, we would like to express our gratitude to Reviewer #1 for providing us with comments and suggestions to help improve the quality of our paper. We have answered your comments point-by-point below and have marked our changes in red in the new version of the manuscript.

Reviewer #1: 

Major revision: My prior comment concerning using the appropriate statistical testing method based on the distribution of the data has not been adequately addressed. The sample sizes are too small to test for normal distributions; therefore, nonparametric tests are called for. 

Prior Comment: A comprehensive reanalysis is required. 

Since sample sizes of 5 and 6 are too small to be tested for a normal distribution, use nonparametric tests for the analysis, i.e., use the Wilcoxon rank sum test instead of the t-test. Furthermore, summarize data using medians and first and third quartiles instead of means and standard deviations. 

A: We thank the reviewer for the comment. Following the recommendation, we have implemented a new non-parametric analysis and modified the entire document accordingly.

Minor revision: 

Line 101: For the power justification, indicate if the t-test was the test used for the "comparisons of means test". 

A: Thank you for the comment. We have added the information. As the reviewer points out, a t-test is the test used.

 

A: First of all, we would like to express our gratitude to Reviewer #2 for providing us with comments and suggestions to help improve the quality of our paper. We have answered your comments point-by-point below and have marked our changes in red in the new version of the manuscript.

Reviewer #2: 

The text calls out Figure 1 to represent study flow but the attached figure 1 is the CONSORT Diagram. 

A: Modified. 

Please provide the manufacturer information for the bikes used. 

A: Added. 

The figures are still very blurry. The axes and legends cannot be read. 

A: The figures have been updated for better readability.

All other comments have been sufficiently addressed. Some responses to the reviewers should be reflected in the manuscript text and not just in the responses to reviewer comment document. 

A: Thank you very much for the thorough review. We have included the information in the text.

 

A: First of all, we would like to express our gratitude to Reviewer #3 for providing us with comments and suggestions to help improve the quality of our paper. We have answered your comments point-by-point below and have marked our changes in red in the new version of the manuscript.

Reviewer #3: 

Thank you to the authors for the answers. The protocol used is correct, and the overall explanation of the findings is clear. However, I suggest checking these two articles for further improvement of the FTP concept. 

Wong S, Burnley M, Mauger A, Fenghua S, Hopker J. Functional threshold power is not a valid marker of the maximal metabolic steady state. J Sports Sci. 2022 Dec;40(23):2578-2584. doi: 10.1080/02640414.2023.2176045. Epub 2023 Feb 20. PMID: 36803419. 

Vinetti G, Rossi H, Bruseghini P, Corti M, Ferretti G, Piva S, Taboni A, Fagoni N. Functional Threshold Power Field Test Exceeds Laboratory Performance in Junior Road Cyclists. J Strength Cond Res. 2023 Sep 1;37(9):1815-1820. doi: 10.1519/JSC.0000000000004471. Epub 2023 Feb 2. PMID: 36692223; PMCID: PMC10448799.

A: Thank you very much to the reviewer for their comment. We have carefully read the articles you indicated. For the next study, we will take into account the aspects related to FTP mentioned in the provided studies.

---

## [Decision Letter · Decision Letter 2]

22 Jul 2024

PONE-D-24-04397R2Effect of high-dose β-Alanine supplementation on uphill cycling performance in World Tour cyclists: A randomised controlled trial.PLOS ONE

Dear Dr. López-Román,

Thank you for submitting your manuscript to PLOS ONE. After careful consideration, we feel that it has merit but does not fully meet PLOS ONE’s publication criteria as it currently stands. Therefore, we invite you to submit a revised version of the manuscript that addresses the points raised during the review process.

We look forward to receiving your revised manuscript.

Kind regards,

Fenghua Sun

Academic Editor

PLOS ONE

Journal Requirements:

Reviewers' comments:

Reviewer's Responses to Questions

**Comments to the Author**

1. If the authors have adequately addressed your comments raised in a previous round of review and you feel that this manuscript is now acceptable for publication, you may indicate that here to bypass the “Comments to the Author” section, enter your conflict of interest statement in the “Confidential to Editor” section, and submit your "Accept" recommendation.

Reviewer #1: (No Response)

Reviewer #2: All comments have been addressed

Reviewer #3: All comments have been addressed

2. Is the manuscript technically sound, and do the data support the conclusions?

Reviewer #1: Yes

Reviewer #2: Yes

Reviewer #3: Partly

3. Has the statistical analysis been performed appropriately and rigorously? 

Reviewer #1: Yes

Reviewer #2: Yes

Reviewer #3: I Don't Know

4. Have the authors made all data underlying the findings in their manuscript fully available?

Reviewer #1: Yes

Reviewer #2: Yes

Reviewer #3: Yes

5. Is the manuscript presented in an intelligible fashion and written in standard English?

Reviewer #1: Yes

Reviewer #2: Yes

Reviewer #3: Yes

6. Review Comments to the Author

Reviewer #1: Suggested minor revisions:

1- The standard statistical term for average is mean.

2-Table 1: Summarize the subject characteristics using median and IQR since the sample size is small.

Reviewer #2: Thank you for addressing the remaining concerns. The authors have adequately addressed concerns. I have no additional comments to provide the authors.

Reviewer #3: I can state that the authors have adequately addressed the comments I raised in the previous round of review.

7. PLOS authors have the option to publish the peer review history of their article (what does this mean?). If published, this will include your full peer review and any attached files.

Reviewer #1: No

Reviewer #2: No

Reviewer #3: No

---

## [Author Response · Author response to Decision Letter 2]

24 Jul 2024

AUTHORS REPLY

A: First of all, we would like to express our gratitude to Reviewer #1 for providing us with comments and suggestions to help improve the quality of our paper. We have answered your comments point-by-point below and have marked our changes in red in the new version of the manuscript.

Reviewer #1: 

Minor revision: The standard statistical term for average is mean.

A: Corrected.

Table 1: Summarize the subject characteristics using median and IQR since the sample size is small. 

A: Corrected.

---

## [Editor Report · Decision Letter 3]

13 Aug 2024

Effect of high-dose β-Alanine supplementation on uphill cycling performance in World Tour cyclists: A randomised controlled trial.

PONE-D-24-04397R3

Dear Dr. López-Román,

We’re pleased to inform you that your manuscript has been judged scientifically suitable for publication and will be formally accepted for publication once it meets all outstanding technical requirements.

Kind regards,

Fenghua Sun

Academic Editor

PLOS ONE
---

## [Editor Report · Acceptance letter]

23 Aug 2024

PONE-D-24-04397R3 

PLOS ONE

Dear Dr. López-Román, 

I'm pleased to inform you that your manuscript has been deemed suitable for publication in PLOS ONE. Congratulations! Your manuscript is now being handed over to our production team.

Kind regards, 

on behalf of

Dr. Fenghua Sun 

Academic Editor

PLOS ONE